# Community Pharmacy Services in Malanje City, Angola: A Survey of Practices, Facilities, Equipment, and Staff

**DOI:** 10.3390/pharmacy10020035

**Published:** 2022-02-22

**Authors:** Bernardo Nicodemo Chimbuco, Mateus Alfredo Ferreira, Euclides Nenga Manuel Sacomboio, Eduardo Ekundi-Valentim

**Affiliations:** 1Department of Health Sciences, Polytechnic Institute, University Rainha Nginga a Mbande, Malanje 0000251, Angola; bernardo.chimbuco@uninjingambande.ed.ao (B.N.C.); mateusalfredo8@gmail.com (M.A.F.); 2Department of Scientific Research and Postgraduate Studies, Institute of Health Sciences, Agostinho Neto University, Luanda 0000222, Angola; euclissacomboio@gmail.com

**Keywords:** public health, human resources, infrastructure, Angola. LMIC

## Abstract

A community pharmacy, also known as a retail pharmacy, is the most common type of pharmacy that allows the public access to their medications and advice about their health. The conditions existing in the community pharmacy, as well as the qualification of the staff who work there, are fundamental for the compliance of good pharmacy practices. Objective: To assess the practices, facilities, equipment, and personnel of community pharmacies in the Municipality of Malanje. Methods: A cross-sectional observational study with a quantitative and qualitative approach. Through a simple random sampling technique, 20 pharmacies were selected from a universe of 73 reported by official authorities. Results: no pharmacist was acting in the local pharmacies, and their activity was supported by other professionals, particularly intermediate nursing technicians (57%). Most pharmacies were in the peri-urban area, and their functional areas, equipment, and utilities were not in accordance with Angolan law. In addition, the distribution of some drugs that are not over-the-counter was observed. Conclusion: community pharmacies in Malanje develop their activity in disregard of the law, constituting a considerable weakness that affects the observance of pharmacy service standards.

## 1. Introduction

A community pharmacy is a pharmacy that deals directly with people in the local area, and its main responsibilities include compounding, counseling, and dispensing of drugs to the patients, accuracy, and legality along with the proper procurement, storage, dispensing and documentation of medicines [1]. Community pharmacies have their activity strongly regulated through specific legislation. The legislation of the Angolan pharmaceutical sector has gone through many phases. In 2000, the National Directorate of Medicines and Equipment (DNME) was established as the regulatory authority for the pharmaceutical sector [2]. At the time, this entity was charged with drafting standards for the promotion, production, use, and maintenance of health technologies in the field of medicines, medical and surgical means, and equipment. However, responsibility for the licensing of pharmaceutical establishments was only given to it in 2020 [3]. 

A year later the DNME was extinguished, and the Medicines and Equipment Regulatory Agency (ARMED) was created [4]. Among its attributions, it will cooperate with national, regional, and international entities in the fight against counterfeiting and smuggling of medicines; registration of medicines and medicinal plants for introduction in the national market; elaboration of the National List of Essential Medicines; National Medicines Form; and Therapeutic Index and Angolan Pharmacopeia, the licensing and supervision of pharmaceutical activity [4]. With the creation of ARMED, the country now has a robust structure that is expected to put in order and give dynamism to all processes related to medicines and health technologies, with particular emphasis on community pharmacies that have grown in alarming numbers, many without the human resources and minimum infrastructure recommended by law. The president of the National Association of the Pharmaceutical Industry (ANIF), said that despite the optimism about the objectives of creating the regulator agency (ARMED), the problem that urgently needs to be solved is the lack of regulation of the Angolan pharmaceutical market, which has led to uncontrolled prices of medicines sold throughout the country [5].

The Angolan legislation establishes that community pharmacies must have the following areas with its minimum’s measures: a room for public attendance (30 m^2)^, verification area (17 m^2^), an office (8 m^2^), sanitary facilities for the staff (3.5 m^2^), warehouse (20 m^2^), running water and sewage. The mandatory types of equipment are a worktable covered with slate, marble or stainless steel, a glazed cabinet for storing exposed medicines in the distribution room, a clothes closet for storage of the pharmacy staff’s outerwear, refrigerator, air conditioners and thermo/hygrometer. About the staff, this law recommends that only the pharmacist is responsible for all information relating to medicines dispensed to the public among pharmacy employees, and no pharmacy can run without this professional [6]. 

Also based on the aforementioned law, pharmacies are classified into three different levels depending on the services they provide to the population. Pharmacies of the first level are those that in addition to medicines and health products, are dedicated to the galenic production and preparation of products for external use, or the realization of clinical, bromatological or toxicological analytical tests; pharmacies of the second level are those engaged in the sale of medicines, medical and surgical materials, products for external use and the performance of clinical analytical tests; and pharmacies of the third level are those dedicated to the exclusive sale of essential medicines, products for external use and expendable material [6].

According to the Angolan Ministry of Health, the pharmaceutical career pertains to the category of diagnostic and therapeutic technicians. There are professionals in the pharmaceutical area with the basic level, secondary level, and degree. However, the first has already been discontinued in terms of training, although there are some still active [3]. Data from the Angolan National Direction of drugs and equipment, there are approximately 509 pharmacists in a country with more than 30 million inhabitants [7], corresponding to approximately 1.7/100,000, very close to Namibia 1/100,000 [8], and Tanzania 2/100,000 [9]. They are, however, below the ratio of pharmacists per inhabitant in other African countries, such as South Africa 27/100,000 [10], Botswana 6.5/100,000 [11], Zambia 7/100,000 [12], and Kenya 5/100,000 [13], demonstrating a severe shortage of trained pharmacists. 

Higher education in the pharmaceutical field is very recent, and the first course was established in 2001 in the country’s capital [14], and there is not a fair distribution of the pharmacist around the country. Data from the Ministry of Higher Education indicate the existence of 12 undergraduate courses in pharmacy, of which ten are in private institutions and two in public ones [15].

The lack of pharmacists in Africa is a reality, and in recognition of the shortage of pharmacists, in some countries such as Ghana, the Board of Pharmacy licenses drug sellers and provides them with some training [16]. In Angola, the available data do not show any specific intervention by health authorities to mitigate the shortage of pharmacists. However, what we have seen in the daily routine is not different from Owusu-Daaku’s description in Ghana: outlets in informal markets and even street vendors dispensing medicines with other products, often without training in the field, and serving as important sources of medicines in many communities [17]. 

Africa has been transformed into a continent of opportunity for pharmacists and patients based on the continued growth of the African pharmaceutical industry, the shortage in the number of trained pharmacists, increasing patient access to drugs previously unavailable on the continent, the rise of large cities, the expansion of healthcare capacity, and the maturation of the business environment, Holt (2015) cited by Sanches [14].

Despite the growth of the pharmaceutical industry in Africa reported above, Angola has no pharmaceutical manufacturing facilities. The country is dependent on imports in the public and private sectors, as well as donations from the international community. To import pharmaceutical products, Presidential Decree N. 180/10 establishes the need for authorization from the regulatory authority of the pharmaceutical sector, customs control, in addition to the General Inspectorate of Health and other inspection entities [18].

One of the main points of intersection among the pharmacist, the patient and the industry is in the process of dispensing the medication, which must be done by observing good pharmacy practices. The recommendations of the International Federation of Pharmacy’s working group for the implementation of good pharmacy practice is access to pharmacy services and personnel [19].

Angola is a country whose lack of data in world statistics is recurrent, for example, unlike African countries like Namibia, South Africa, Nigeria, Zimbabwe, Mozambique, just to name a few, the country does not have data in the WHO reports on the pharmaceutical workforce [13], which should be a matter of concern for the Angolan health authorities. 

Also, the availability of pharmacists and infrastructure to provide pharmaceutical services based on good pharmacy care practices remains an unknown field. This study aimed to describe the community pharmacies in the city of Malanje, focusing on staff, facilities and equipment, to propose actions that can be implemented to improve the quality of services, and consequently the safety of the population.

## 2. Methodology

### 2.1. Type of Study

It was a cross-sectional descriptive study with a quantitative and qualitative approach. 

### 2.2. Place of Study

The study was carried out in the Municipality of Malanje, province of Malanje. It has a territorial extension of 2422 Km^2^, and an estimated population of 221,785 inhabitants. Geographically it is limited by the following cities: North Kiuaba Nzongi, East Mucari, South Cangandala and Mussende and West Kalandula and Cacuso. Nine neighbourhoods comprise the administrative division, namely Maxinde, Ritondo, Cangambo, Campo de Aviação, Catepa, Vila Matilde, Quizanga, Cula Muxito, and Vuanvuala. It has a dry tropical climate, and the temperature varies from 20–25 degrees centigrade.

### 2.3. Population and Sample

The population was made up of the total number of community pharmacies legally controlled by the Malanje Provincial Health Office. This office provided the list, contacts, and addresses of community pharmacies of the municipality. In the list, there were 73 pharmacies, and of these, 20 pharmacies were selected using the simple random probability sampling technique, the lottery method.

### 2.4. Ethical Procedures

The Ethics Committee of the Polytechnic Institute of Malanje approved the project, registered under the N.231/18. A letter was then sent to the Provincial Health Office requesting a list of all legal community pharmacies. In response to this list, and after selecting the pharmacies, letters were sent to them requesting their consent to carry out the survey. Following approval, the pharmacy’s technical manager was interviewed and a consent form was signed, ensuring that the data obtained are for scientific purposes only. The name, image of the institution, and the participants would be kept confidential, and they were free to withdraw from the study if they deemed it necessary.

### 2.5. Data Collection Procedures

The field study was carried out in the second semester of 2018, using interview and observation techniques. The interview consisted of applying a previously prepared questionnaire, with questions to characterise the infrastructure, human resources, and operational aspects of pharmacies. The observation technique was used to check the organisation and packaging of medicines in the dispensing room and the warehouse, as well as how the drugs were dispensed to the public.

Initially, information related to identification data of the pharmacies such as name, legal form (private or public), geographical location, classification according to the services they provided to the public was collected. Data regarding the infrastructure, equipment’s, and utilities were obtained.

### 2.6. Analysis and Data Processing

After the collection, for quantitative analysis, the data was by descriptive statistics (frequency, percentage) using SPSS v20.0 (IBM SPSS Statistics, Chicago, IL, USA). The selective observation was used to check functional aspects of pharmacies. 

## 3. Results

Of the 20 pharmacies studied, none of them had a pharmacist; however, 11% of employees with some technical training in the pharmaceutical area were reported, between basic and intermediate technicians. Other data attracting attention are the presence of employees with training in the area of education working in pharmacies (Table 1). 

One of the pharmacies had more than 10 employees, and the male gender was predominant (Table 2).

About the utilities of the pharmacies, 100% had electricity while only 15% had running water and sewage connected to the public network. According to the location area, 11/20 (55%) were in the urban area. As for furniture and equipment, all did not have the equipment established by national legislation (Table 3).

The pharmacies had the minimum functional areas to perform their activity, but besides the absence of a laboratory for testing and a galenical area, the existence of a private area, medicine reception room and break room were not common, as shown in Table 4.

According to the observational data, most pharmaceutical establishments had specific opening and closing times, i.e., they did not operate permanently (24/7). Medicines were not organised in alphabetical order, pharmaceutical form, and pharmacological group. The storage areas did not have equipment such as steel cupboards with keys for storing controlled products, which were kept together with over-the-counter medicines. They did not have refrigerators for the storing of heat-sensitive products. They lacked frames for fixing medicine boxes and other pharmaceutical inputs. Therefore, medicine boxes were disposed of directly on the floor, and none of them had a control system for temperature and humidity (thermo/hygrometers). Additionally, was observed the dispensation of some medicine that are not over-the-counter without a prescription.

## 4. Discussion

The present survey on practices, facilities, equipment, and personnel of community pharmacies in Malanje brings to light the conditions under which such activity is developed, as well as the actors involved. The data point to a lack of pharmacists working in the community pharmacies in the Municipality of Malanje, which constitutes a weakness in the health system, and a danger to the health of the community. In developing countries, where health needs are very high and public sector health care provision is limited, pharmacists are seen as being well placed to provide advice, on the management of common symptoms and long-term conditions [16], and could make a considerable contribution to healthcare delivery [20].

Although the data are only from the survey conducted in 20 pharmacies, we can consider them to be in line with the data from the National Directorate of Drugs and Equipment (DNME), which reported the existence of 509 pharmacists throughout the country, and this number is far behind the WHO recommendation (1:2000); the lack of pharmacists in Malanje is a consequence of the lack of these professionals in the Angolan market, and this reality not only in Angola, it is a characteristic of underdeveloped and developing countries [8,9,10,11,12,13]. In the same vein, in Ghana 619 pharmacists are serving 2.9 million people in Greater Accra [21], and there are 8102 pharmacists for more than 160 million inhabitants in Pakistan, according to the Report of the Health System Review Mission, cited by Azhar and coworkers [22], demonstrating a severe shortage of trained pharmacists in developing countries. The available figures show that the shortage of pharmacists in European countries is not comparable to the situation in Africa. Data from 2017 indicated Belgium as the best-placed European country in the ratio of pharmacists per inhabitant, which was 119 per 100,000; and the Netherlands as the worst placed in this ratio, which was 21 pharmacists per 100,000 [22]. Still, the last European country in the ratio was only below South Africa, which is the best-placed African country [10]. The Global Pharmacy Workforce report in which the years 2006, 2009 and 2012 were analyzed shows that pharmacy workforce density varied considerably across WHO countries and regions and generally correlated with population numbers and economic development indicators at the national level [23]. Countries and territories with lower economic indicators tended to have relatively fewer pharmacists and pharmacy technicians [9].

If on the one hand pharmacy training in Angola is very recent [14], and there are so far 12 undergraduate courses [15], the lack of pharmacists in Malanje may also be indicative of the poor geographical distribution of pharmacists in the country. Malanje is located 400 kilometers from the capital Luanda and, in general, the interior provinces are not very attractive. These findings are consistent with data from Ethiopia [24] showing that of the 1898 pharmacists, about half (45.9%) worked in the country’s capital, Addis Ababa. In the light of the absence of pharmacists, pharmacies resort to other professionals, such as doctors, nurses, intermediate nursing technicians, and even professionals in the field of education. The presence of health professionals in the process of dispensing medicines is a practice that is observed in other parts of the world, depending on the specific legislation of each country. For example, in Malaysia, doctors dispense medications as part of their professional practice [25]. However, the Angolan legislation determines that no pharmacy can operate without a pharmacist technical manager, and this is one of the requirements to be observed in the licensing process [6]. But the data show that they are working even though there is no pharmacist.

In Malanje’s pharmacies the functional areas, equipment, conditions, and utilities are not in compliance with the universal standard and the law. For instance, 85% of establishments lack running water and networked sewage, and 15% lack glass cabinets for medicines in the distribution room and product display. Water and sewage are fundamental elements for the health of the environment. Their absence constitutes an attack on public health since it can raise questions about the maintenance of products in hygienic conditions and so on.

Regarding the problem of functional areas in community pharmacies, in some countries, such as Brazil, in the face of the non-conformities found, a model was proposed that was accepted by the regulatory agency, the National Health Surveillance Agency [26]. The WHO in 2018 in Mozambique also proposed a plant model for pharmacies [27].

Pharmacy practice models in developing countries vary significantly from one country to another [23], and pharmaceutical services in these countries face some specific challenges unlike those faced by pharmacists in the developed world [23]. In most developing countries, the lack of appropriate and good-quality medicines is the most common problem encountered [28], as well as the lack of infrastructure with appropriate functional areas and utilities, as evidenced by these results. 

The findings of the present study show that all community pharmacies belonged to the private sector, none were open 24/7, they were located primarily in peri-urban areas (55%) and only offered dispensing medicines. On this subject, we found several studies with different findings. For example, a survey carried out in Brazil by Bareta showed that 68% of pharmaceutical establishments were private [29]; and in South Africa, a study carried out by the South African Pharmacy Council (SAPC) identified 63% of private community pharmacies [30]. In Angola it seems to be a generalized tendency, the only pharmacies under state responsibility are the hospital ones, leaving more space for the private sector to explore the community pharmacy market.

Another aspect to be highlighted is the non-performance of laboratory tests in pharmacies; despite this being in the law, in practice, it is not observed and the infrastructures themselves are not adapted for this purpose, lack of running water for example, besides the possible lack of equipment and qualified human resources. Rapid tests would be the most appropriate to perform in pharmacies, and there seems to be an openness of the Ministry of Health in this direction. Recently they published an announcement inviting pharmacies that meet certain criteria to apply to perform rapid tests for COVID-19.

It was observed in terms of the delivery of medicines that, even though they were not over-the-counter, they were dispensed without the presentation of a medical prescription, and patients were neither guided nor accompanied; the drugs were simply handed out like pieces of cake in a bakery or candy in a shopping mall. Advice-giving and recommending appropriate medicines are fundamental to community pharmacy activity, however, with no pharmacist present, pharmacies are unable to comply with the recommendations of good pharmacy practices. In developing countries, many products are requested and supplied without a prescription. In these instances, the good pharmaceutical practice would require that the pharmacy staff ensure appropriate products are supplied with relevant labelling and advice [16]. As the local community pharmacies are private, their incomes are derived principally through the sale of medicine and other products. Potential conflicts between the business and professional interests of pharmacists are frequently highlighted [16,31].

The mission of pharmacy practice is to provide medicines and other health products and services to help people and society make the best use of them [32]; however, without pharmacists, this mission is compromised. According to Cipolle et al., 1998, cited by Droege [33,34], Good Pharmacy Practice is the way to implement pharmaceutical care, and its response to the patient’s medication-related needs in a comprehensive, and rational manner through a timed scheme of tasks, in which the practitioner ensures that drug therapy is appropriately indicated, effective, safe, convenient, and consistent with existing practices in the health care system. Besides saving thousands of lives, pharmacists’ interventions also help save large sums of money for health care [35].

In Angola, although the law already establishes the requirements and procedures, these results raise questions about how local health authorities have been licensing community pharmacies and reinforce the urgent need to meet the requirements established in the law throughout the licensing process, thus avoiding situations such as the lack of pharmacists, and precarious infrastructure, making it impossible to comply with the recommendations of the International Federation of Pharmacies working group for the implementation of Good Pharmacy Practices: access to services and pharmaceutical staff [19].

We think that with the new Regulatory Agency, the activities of the pharmaceutical sector will be subject to better control and supervision. The agency is responsible for licensing and supervising the exercise of pharmaceutical activity. In addition, it controls the consumption and use of medicines and health technology, as well as ensuring their rational use, and pronounces on the construction, rehabilitation, equipment, and operation of pharmaceutical establishments [4]. Angolan society is eager to see how the creation of the new regulatory agency will improve the quality of service offered by pharmacies.

The study highlighted the need for greater rigor in the licensing of pharmacies. In addition, it demonstrated the urgent need to implement regular training, updating and refresher programs for professionals who provide care in community pharmacies. More monitoring visits, help and control should be performed by the competent authorities, to identify and help improve the existing non-conformities; and those that do not adjust within a given time should lose their licenses to operate and consequently be closed. Meanwhile, there is an undeniable need for pharmaceutical assistance to be provided by professionals trained in the area, because only they have the necessary skills to develop such activities with the necessary quality, so the training of more pharmacists in Malanje province, and Angola in general, is the best way to solve the problems identified.

To the best of our knowledge, this is the first survey of community pharmacies in one locality in Angola. The study involved only community pharmacies in Malanje in a specific period, and the findings may not be generalizable to pharmacies in other locations in the country. The picture we obtained encourages us to expand the study to other cities, and in this way obtain data that better reflect the country’s reality.

## 5. Conclusions

No pharmacists are working in community pharmacies services in Malanje, and the functional areas, equipment, and utilities are not in compliance with Angolan law. All these non-compliances constitute considerable weaknesses that impact the dispensing, labelling, instruction of patients, records, health information, patient counselling, and pharmaceutical care, which are all standards of pharmacy services, thereby posing a risk to patient safety. 

## Figures and Tables

**Table 1 pharmacy-10-00035-t001:** Distribution of professional qualifications of community pharmacies staff in the Municipality of Malanje, second semester of 2018.

Professional Qualification	Freq.	%
Physician	2	2
Nursing	2	2
Graduate in Education	1	1
Intermediate Pharmacy Technician	11	10
Intermediate Nursing Technician	64	57
Intermediate Education Technician	2	2
Basic Pharmacy Technician	1	1
Basic Nursing Technician	30	27

**Table 2 pharmacy-10-00035-t002:** Distribution of staff and gender in the community Pharmacies in the Municipality of Malanje, second semester of 2018.

Staff Number	Pharmacies		Sex				Total	
			Men		Women			
	Freq.	%	Freq.	%	Freq.	%	Freq.	%
1–4	13	65	31	27	21	19	52	46
5–9	6	30	32	28	18	16	50	44
≥10	1	5	7	6	4	4	11	10
Total	20	100	70	61	43	39	113	100

**Table 3 pharmacy-10-00035-t003:** Distribution of equipment of community pharmacies of the Municipality of Malanje, second semester of 2018.

Equipment	Freq	%
Worktable covered with slate	4	20
Refrigerator	7	35
Glazed cabinet for medicine storage exposed in the distribution room	3	15
Clothing locker for clothing for outside use by pharmacy staff	2	10
Waiting seats	11	55
Product Showcase	3	15
Air conditioning system	17	85
Computers with installed management software	7	35

**Table 4 pharmacy-10-00035-t004:** Distribution of the community pharmacies functional areas in the Municipality of Malanje, second semester of 2018.

Functional Areas	Freq	%
Public Service or Dispensing area	20	100
Sanitary facilities for technical and auxiliary staff	20	100
Offices	19	95
Warehouse	19	95
Private service area	2	10
Medication reception room	1	5
Break room	6	30

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
