# Peer review of "Community Pharmacy Services in Malanje City, Angola: A Survey of Practices, Facilities, Equipment, and Staff"

_pharmacy, 2022, doi:10.3390/pharmacy10020035_

Round 1
Reviewer 1 Report
The authors are attempting to raise awareness to the many problems of providing pharmacy services in Malanje City, Angola. They conducted a cross-sectional survey of 20 randomly selected community pharmacies to determine the conditions under which pharmacy care is rendered. While I admire the authors' attempt to characterize pharmacy services and the many infrastructure, human resource, and equipment-related deficiencies they suspected, the manuscript is poorly written, lacks focus, and needs re-written entirely in order to be reconsidered for publication. I point out examples of issues with the manuscript.
Title: vague. Consider "Community pharmacy services in Malanje City, Angola: A survey of practices, facilities, equipment, and personnel"
Abstract:: the authors state that they use quantitative and qualitative approaches, but I could not find any qualitative data or analysis in the manuscript. I do not know what a "medium education technician" is.
Key words: Add "Angola" and "LMIC". Change community pharmacies to "community pharmacy"
Introduction: the authors try to summarize the current state of pharmacy practice in other parts of the world, including low and middle income countries (LMIC), and why they performed the survey. At line 84, "the availability of staff and infrastructure to provide pharmaceutical services based on good pharmacy care practices is unknown instead of "remains an undiscovered field."
Methodology: why were 20 pharmacies randomly selected? On what basis and why? Why not survey all 73 pharmacies? How were they contacted? Were the pharmacy personnel interviewed in person? Where is the questionnaire that was used? What does it mean that "observation data were analysed from a qualitative point of view?"
Results: what does this statement mean? "09/20 (45%) were of the second level of pharmaceutical service". At line 159, I wouldn't use compartments to describe facilities. Functional areas is a better way to describe these rooms. At line 160, instead of "technical conditions" I would use "utilities" to describe electricity, water, and sewage. The Figures need to be re-labeled as Tables. Figure 2 needs to be expanded to make it easier to read. At line 183, 70/100 is 70%, not 61% as the authors stated.
Figures are called tables in the text.
References are not in mdpi style. Reference #35 is incorrect. Many are incomplete.
Author Response
Dear reviewer
The authors would like to thank you for your patience, quality, and depth of analysis in reviewing this manuscript, which has greatly enhanced its quality and the information to be shared with the scientific community. We hope that we have resolved all inconsistencies found in the manuscript.
Please see the attachment.

Reviewer 2 Report
Please see attached file

Author Response

(The authors gave the same response as above.)

Round 2
Reviewer 1 Report
The authors have addressed most of the reviewers' comments and suggestions for improvement, and I believe that the paper is much improved and more focused on the critical problems facing Angolan pharmacy.
Author Response
Dear Reviewer,
Thank you very much for the rich suggestions that have significantly improved the quality of our work.
Our sincere thanks
Reviewer 2 Report
Dear Authors,
Thank you for your revised manuscript. The introduction is so much richer and clearer now, and it is much easier to see what you want to tell in the paper.
I would like to see the introduction ending with a clear statement of the aim of the paper, like: “The aim of this project is…… (to describe the pharmacy services in Malanje City, Angola and propose actions… or something like that). When you have a clear aim, the conclusion has to answer this.
And just a few minor questions:
Line 260: what does the number (191/10) mean? Is it needed here? I see later in the chapter that this is reference 6.
Line 266: Does the law actually just “recommend” or does it “request”? (is it compulsory or mandatory?)
Line 270: please choose “level” or “class” – not both.
Lines 278-9: What does this mean?
Lines 282-7: 509/30 mio equals approximately 1.7/100.000 which is something between Namibia and Tanzania thus not “far below all other African countries”.
Lines 302-6: What do you mean to say here?
Line 545: Is there a special guideline for “good pharmacy management practice” or should it just be “good pharmacy practice”?
Line 580: I suppose the CEISPM is the abbreviation for the ethics committee, but again – what do the numbers 231/18 refer to? Is it a reference number for your specific approval? Or what is it?
Lines 678-9: Method for the qualitative analysis?
Table 2: I struggle to read this table. What does it tell me? I tried this interpretation: 7 men and 4 women (11 in all) worked in community pharmacies with 10 or more employees. This must indicate that there was only one pharmacy with 10 or more employees – right? But I can’t figure out how many pharmacies had 1-4 or 5-9 employees. Please clarify.
Table 2: it has to be “≥10” – otherwise the number “10” is not included in any category.
Lines 839-40: You need a “full stop” after “electricity – as the sentence is now, it is confusing. Should be: “About the utilities of the pharmacies, 100% had electricity. Running water and …….” Or why not: “About the utilities of the pharmacies, 100% had electricity while only 15% had running water and sewage connected to the public network”?
Table 4: “Cafeteria” indicates that you can buy food there – I suppose you mean a separate room for eating/taking coffee breaks? Could be called a “break room”.
Line 878: I suppose you mean refrigerators for “storing” rather than “packaging”.
Lines 881-2: please rephrase.
Lines 1145-7: you have already given those numbers earlier – can you avoid repeating?
Lines 1173-5: You state that the legislation determines that no pharmacy can operate without a pharmacist – but they do!
Lines 1347-9: Hospital pharmacies being the only government owned pharmacies is quite common all over the world as far as I know.
Lines 1358-9: Please read this again and clarify what you mean. OTC medicines require no prescription (that is the definition of OTC), so this is a bit confusing.
Lines 1360-1: Yes, giving advice is essential, but is requires a pharmacist, which they don’t have, so what should they do?
Lines 1370-1: Yes, the local authorities have failed and that is the big point here.
Conclusion: Agree fully and in addition this is a risk to patient safety.
Patient safety is the main point for all the rules and requirements and for Good Pharmacy Practice. Maybe you could go a bit into this theme in the paper as well – or at least mention it because it is so essential.
You have discovered a problem, but what can be done to solve it? Better control with licensing (as you suggest in the discussion)? Do you know any examples of this from other areas? Developing higher education to educate more pharmacists? Other ideas? This could also be discussed.
Author Response
Dear Reviewer,
Please see the attachment.
Best regards,
Eduardo
